

# In-vehicle network intrusion detection systems: a systematic survey of deep learning-based approaches

Feng Luo, Jiajia Wang, Xuan Zhang, Yifan Jiang, Zhihao Li and Cheng Luo

School of Automotive Studies, Tongji University, Shanghai, China

## ABSTRACT

Developments in connected and autonomous vehicle technologies provide drivers with many convenience and safety benefits. Unfortunately, as connectivity and complexity within vehicles increase, more entry points or interfaces that may directly or indirectly access in-vehicle networks (IVNs) have been introduced, causing a massive rise in security risks. An intrusion detection system (IDS) is a practical method for controlling malicious attacks while guaranteeing real-time communication. Regarding the ever-evolving security attacks on IVNs, researchers have paid more attention to employing deep learning-based techniques to deal with privacy concerns and security threats in the IDS domain. Therefore, this article comprehensively reviews all existing deep IDS approaches on in-vehicle networks and conducts fine-grained classification based on applied deep network architecture. It investigates how deep-learning techniques are utilized to implement different IDS models for better performance and describe their possible contributions and limitations. Further compares and discusses the studied schemes concerning different facets, including input data strategy, benchmark datasets, classification technique, and evaluation criteria. Furthermore, the usage preferences of deep learning in IDS, the influence of the dataset, and the selection of feature segments are discussed to illuminate the main potential properties for designing. Finally, possible research directions for follow-up studies are provided.

# INTRODUCTION

Technological developments in the automotive sector are promoting the emergence of innovative service paradigms like the lane-keeping assist system (LKAS) or cruise control system (CCS). These convenience-adding functions have become indispensable parts of improving driving and a better customer experience, pushing original equipment manufacturers (OEMs) to create technologically advanced innovations supported by connectivity. By 2023, global connected vehicles will jump 134% from 330 million in 2018 to 775 million (*Upstream, 2022*). Unfortunately, the other side of increasing connectivity and complexity is that automotive security risks have become more prominent. Originally, in-vehicle network communication seldom considered cybersecurity risks while more concerned with bandwidth, real-time, and low-cost requirements. Nowadays, an increasing

Corresponding author
Jiajia Wang, 1911047@tongji.edu.cn

number of security researchers demonstrated various vulnerabilities among the different in-vehicle network protocols. As studied by *Miller & Valasek (2015)*; *Petit & Shladover (2015)*; *Sun et al. (2020)*, due to the controller area network (CAN) bus protocol itself lacking proper security support such as authentication and encryption, it is vulnerable to attacks such as denial of service (DoS) and bus injection. While *Talic (2017)* investigates the cybersecurity of automotive Ethernet and its accompanying protocols, pointing out that the attacks against it are real. The expanded attack surface enables hackers to find new attack vectors and exploit every vulnerability to compromise vehicles and networks.

Given the long life cycle of vehicles and the existing various attack vectors against IVN, designing and conducting an utterly secure IVN system in the development stage is difficult (*Wu et al., 2020*). IVN IDSs provide persistent protection by monitoring and analyzing network traffic. Compared with other countermeasures such as encryption (*Farag, 2017*), authentication (*Nilsson, Larson & Jonsson, 2008*) and hardware-enforced isolation (*Hu et al., 2020*), IVN IDSs have significant advantages: (1) they do not generate extra IVN traffic, (2) nor modify communication protocol or occupy the payload, and (3) they are incapable of affecting real-time performance. Additionally, UNECE WP.29 establishes clear performance requirements for OEMs and suppliers to deal with risks, including monitoring and detecting cybersecurity events. Therefore, it is essential to employ IDS to monitor and analyze network traffic during the entire life cycle, revealing potentially suspicious activities and further providing evidence to prevent adversaries from threatening stakeholders.

Many researchers focus on developing and deploying IDS using various machine learning techniques, intending to automatically identify intrusion events in the IVN context in recent years. With the increase of more communication nodes and new service paradigms, massive and multi-dimensional traffic data are generated every moment, and attack vectors are growing diversified and sophisticated, making the methods based on traditional machine learning such as hidden Markov models (*Narayanan, Mittal & Joshi, 2016*), support vector machine (*Avatefipour et al., 2019*) and decision trees (*Tian et al., 2018*) ineffectively deal with the evolving security risks. As part of machine learning techniques, especially variants of artificial neural networks (ANNs), deep learning has been widely incorporated for designing IDSs in different environments. This is because deep learning is capable of performing complex non-linear data transformations to distinguish normal and attack traffic on the network. Deep learning networks are more effective in identifying sophisticated attacks and zero-day attacks (*Khan et al., 2021*) by finding correlations among a mass of training samples without human intervention. Besides, deep learning networks benefit from incremental learning, which enables them to extract new features from training datasets. Deep learning-based IDS schemes proposed by researchers with various network architectures perform feature learning and classification tasks differently. Therefore, a timely systematic literature review of current state-of-the-art studies on IVN IDS using DL methods is necessary.

This article presents a comprehensive survey, taxonomy, and analysis that covers the DL-based papers on research endeavors to detect intrusions in IVN. It begins with a discussion of possible attacks against IVN and briefly outlines publicly available datasets to facilitate understanding their attack implementation strategies as well as the capabilities

of IDS schemes trained with them. It then provides a fine-grained taxonomy of the DL-based IDS schemes that consider the deep network architecture used during the feature selection/extraction or classification stage. Further details the different intrusion detection steps implemented by deep learning approaches and their possible deficiencies and limitations in corresponding subsections. Besides, with the discussion and analysis performed, the survey illustrates the topics that can be further researched.

The main contributions of this survey are as follows:

–Illustrating possible attacks against IVNs and describing attack implementation strategies for commonly used public datasets.

–Proposing a fine-grained taxonomy of studied DL-based intrusion detection schemes to help researchers compare the different approaches in detail. Moreover, the reviewed IDS schemes cover both CAN bus and Automotive Ethernet, and the latter has played an increasingly prominent role in IVNs.

–Comparing some remarkable IDS schemes in the tabular form concerning different facets in input data strategy, benchmark datasets, classification technique, and evaluation criteria.

–Discussing possible future research directions that can be further investigated to improve the performance of DL-based IDS schemes.

The remainder of this survey is organized as follows. 'Previous Surveys' reviews and compares the previous surveys related to in-vehicle network intrusion detection systems. 'Research Methodology' introduces the research methods for performing this survey, and 'Background Knowledge' briefly describes the possible attacks and datasets against IVN. 'Deep Learning-Based IDS' proposes a fine-grained taxonomy of the investigated IDS schemes and a detailed description of the detection implementation process, followed by a discussion of the obtained findings in 'Discussion'. Section 'Future Research Directions' presents the challenges and future trends, while 'Conclusion' concludes this survey.

## PREVIOUS SURVEYS

In-vehicle network cybersecurity and deep learning technology are mainly researched independently. Only recently, IVN security researchers have turned to deep learning to detect intrusions. There are several related surveys that provide some description of IDS for vehicles. Surveys (*Limbasiya et al., 2022*) mainly focus on analyzing the attack surfaces and corresponding countermeasures. *Dixit et al. (2022)* survey anomaly detection methods in autonomous electric vehicles, but they are not paying attention to in-vehicle networks such as CAN and vehicle Ethernet. Similarly, in *Karopoulos et al. (2022)*, the authors concentrate on analyzing existing classifications of in-vehicle IDS surveys to propose a generic taxonomy. The other surveys are not completely dedicated to deep learning methods and only assign a subsection for a brief description. A detailed comparison of this review with the previous surveys is presented in Table 1, involving (1) the number of DL-based schemes, (2) the focused application areas, and (3) the survey execution method.

This survey differs from other review articles in the following aspects: (i) we followed a systematic literature review process to obtain more comprehensive papers on the IDS

**Table 1  In-vehicle IDS survey comparision.**

| Survey | DL-based schemes[*] | Focused application areas | | Systematic study |
|---|---|---|---|---|
| | | In-vehicle IDS focused | Protocol | |
| *Limbasiya et al. (2022)* | Less | – | | √ |
| *Karopoulos et al. (2022)* | Medium | √ | CAN & Ethernet | – |
| *Dixit et al. (2022)* | Less | – | – | √ |
| *Wu et al. (2020)* | Less | √ | CAN | – |
| *Al-Jarrah et al. (2019)* | Less | √ | CAN | – |
| *Young et al. (2019)* | Less | √ | CAN | – |
| *Lokman, Othman & Abu-Bakar (2019)* | Less | √ | CAN | – |
| This survey | More | √ | CAN & Ethernet | √ |

**Notes.**
*"Less", "Medium" and "More" respectively refer to the number of surveyed articles: 1∼10, 11∼20, >21.

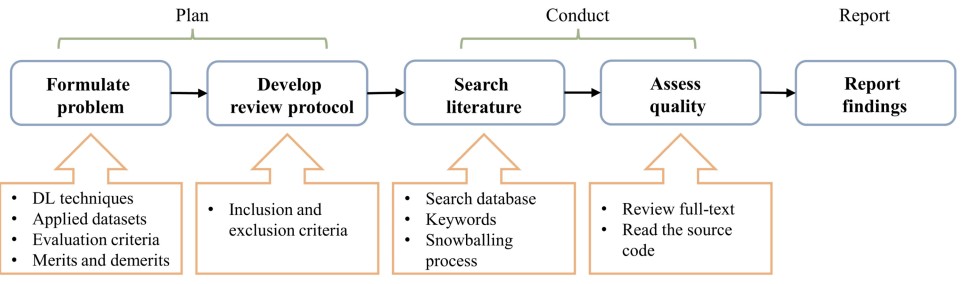

**Figure 1  Research steps.**

designed by deep learning techniques. (ii) Our study covers the most cutting-edge deep learning-based IDS schemes published between 2016 and 2023, which provides more updated information and recent trends where researchers can find potential areas to explore. (iii) We present a fine-grained taxonomy following the neural network architectures to classify the state-of-the-art DL-based in-vehicle IDS schemes. (iv) Our study provides a comparative and critical analysis of the investigated solutions, considering their methods, datasets, and evaluation metrics.

## RESEARCH METHODOLOGY

This study executes a systematic literature review (SLR) of DL-based IDS schemes for securing in-vehicle networks. The procession mainly follows the method demonstrated in *Xiao & Watson (2019)*; *Kitchenham & Brereton (2013)*, which is divided into three stages: planning, conducting, and reporting. The detailed steps are summarized in Fig. 1. 'Planning' introduces the review principles of the article search and selection strategies, and 'Study search and selection' describes the implementation process. Finally, subsequent chapters reveal the findings from the literature review.

**Table 2  Inclusion and exclusion criteria.**

| Criterion | Description |
| --- | --- |
| Inclusion 1: Articles are peer-reviewed. | We want only to analyze studies that have verified the quality and integrity beforehand. |
| Inclusion 2: Works that present methods to detect the anomalies and attacks for in-vehicle networks. | We aim to investigate state-of-the-art solutions for intrusion detection in in-vehicle networks. |
| Inclusion 3: Works are evaluated on datasets. | We want to consider works verified by a comprehensive set of experiments. |
| Inclusion 4: Research published in English. | Considering the readability of papers, we only include English papers. |
| Inclusion 5: Papers published between 2016 and 2023. | We want to obtain the current research state in this area. |
| Exclusion 1: Duplicate works will be excluded. | Articles searched from different databases or snowballing processes generate duplicates. |

## Planning

This section clarifies the review principles, a preset plan specifying the methods used in executing the review. It is beneficial to unify the work behavior of different research participants and repeat this survey for cross-validation.

### Research scope and questions

This survey is dedicated to identifying, classifying, and comparing current DL-based IDS schemes for securing In-vehicle networks. Based on this purpose and the analysis of uncovered content in other reviews, the following Research Questions (RQ) were defined.

RQ1 Which DL techniques(methods) are adopted by each IDS scheme?

RQ2 How do different schemes employ DL techniques in different stages of the intrusion detection procedure to achieve better performance?

RQ3 Which datasets are used for IDS training, validating, and testing purposes?

RQ4 What evaluation criteria are applied to measure the performance of the system?

RQ5 What are the merits and demerits of each IDS scheme for recognizing anomalous behavior in considered domains?

### Inclusion and exclusion criteria

The articles retrieved from databases were further screened to decide whether they should be included for data extraction and analysis. We follow a two-stage strategy: articles were first submitted to a coarsely screened through the article abstract, followed by a full-text review for detailed quality assessment. The inclusion and exclusion criteria used in this survey are listed in Table 2.

## Study search and selection

Firstly, search engines and keywords are determined based on the research question to find the related articles. The articles were searched in the ScienceDirect, IEEE Explore, ACM Digital Library, Springer, Wiley, Hindawi, and MDPI databases due to their ability to cover comprehensive conferences and journals. Then, we searched using the initial keywords *intrusion detection for in-vehicle network* and adjusted the filters. The initial search results include IDS using different approaches like signature-based, entropy-based,

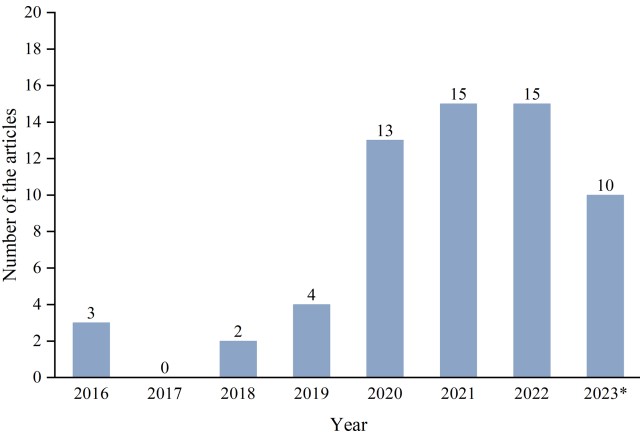

**Figure 2** **Number of works per year.** * Mains only cover the first half of 2023.

parameter monitoring-based, information theory-based, machine learning-based, *etc.* We then redefined our keywords as *intrusion detection*, *anomaly detection*, and *in-vehicle network* with the combination of *deep learning* to obtain more relevant articles. The obtained 507 articles are stored as an initial list. Subsequently, we performed pre-defined inclusion and exclusion criteria, and 51 studies were included. Besides, we performed the snowballing process, including the references of selected articles and the papers that cited selected articles, until no new articles were found. Finally, 62 articles were included in the final list for further analysis.

Figure 2 illustrates the publication year of the IDS schemes investigated in this survey. It can be noted that this survey mainly focuses on the novel IDS methods published since 2016. Since the relatively recent paradigm of in-vehicle network intrusion detection, research works started to rise in this context only from 2019 onwards. It is worth noting that the number of articles is increasing yearly, indicating that this area is still a research focus of in-vehicle network cybersecurity.

# BACKGROUND KNOWLEDGE

This section introduces the possible attacks on in-vehicle networks and provides a detailed analysis of the datasets that are used for training, validating, and testing IDS schemes.

## Introduction of IVN

Modern vehicles consist of numerous electronic control units (ECUs) connected over IVNs, the backbone of vehicles for data transmission among different nodes. There are several communication protocols, *i.e.,* Local Interconnect Network (LIN), Controller Area Network (CAN), FlexRay, Media-Oriented System Transport (MOST), and automotive Ethernet. Among these protocols, CAN typically is in charge of critical real-time data exchanges, while LIN, FlexRay, and MOST mainly play an auxiliary role to the former. Automotive Ethernet, due to its much higher bandwidth and lower cost, is becoming the new major player in this field. Figure 3 shows a typical automotive network topology,

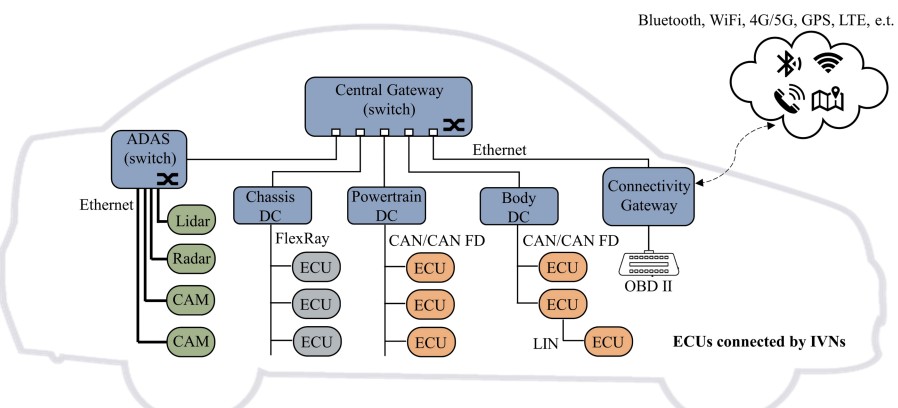

**Figure 3   Typical network topology of a modern vehicle.**

which connects with different types of ECUs, on-board diagnostic port (OBD-II), and Telematics and Infotainment System (TIS). Among investigated IDS schemes, the great majority are intended for CAN and several for Ethernet. Therefore, only CAN and Ethernet are explained in detail in this section.

Currently, the CAN bus has the most widely used from any other in-vehicle communication technology. CAN transmit information by broadcasting its messages, which means that all nodes connected to the CAN bus can actively transmit information to other nodes at any time. Intending to avoid message collisions within communication, it used the arbitration field (CAN ID) of each frame to indicate its priority. And the High-priority packets can always win the bus. The CAN message carries a sequence of payload up to 8 bytes, whereas 64 bytes in CAN Flexible Data (CAN-FD). Typically it transmits periodic varies from 10 ms to 1,000 ms. However, it is vulnerable to malicious attacks due to its broadcastability and lack of native cybersecurity support.

As the amount of data interaction within vehicles increases, specialized automotive Ethernet protocols and services are being developed. Application areas are roughly divided into (1) DoIP (Diagnostic communication over Internet Protocol) based on 100Base-Tx for diagnostic and OTA (Over-the-air) update, (2) AVB (Ethernet Audio/Video Bridging) and SOME/IP (Scalable Service-Oriented Middleware over IP) based on BroadR-Reach, applied in Advanced Driver Assistance Systems (ADAS) and In-Vehicle Infotainment system (IVI), (3) Time Sensitive Network (TSN) based on 1000 Base-T1 for communication backbone. A protocol data unit (PDU) typically encapsulates a header as it passes through the different ISO OSI layers. Following the protocol header is the payload field, which can be from 46 bytes to 1,500 bytes of data. As Ethernet has been the objective standard for interconnecting computers, the enormous experience and tool support base from hacking computer networks might be used to attack vehicles.

## Possible attacks in IVN

In-vehicle networks are vulnerable to several cyberattacks with different severity levels (*Jo & Choi, 2022*; *Kim et al., 2021*). Malicious adversaries can easily attack IVNs due to

the increasingly rich interfaces, such as physical interface (OBD-II (*Koscher et al, 2010*), USB ports (*Jo et al., 2017*; *Mazloom, Rezaeirad & Hunter, 2016*)) and remote interface (WiFi (*Nie, Liu & Du, 2017*), Bluetooth (*Checkoway et al., 2011*), and cellular *Miller & Valasek, 2015*)). The attacks investigated in this paper are based on the assumption that a compromised node exists in the network. The attacker has complete control of the compromised node, with the capacity to transmit forged messages and access the node memory (*Cho & Shin, 2016*). Based on the study of present works, we analyze the most common cyberattacks against IVNs and classify them into the following types, which are helpful for describing attack complexity and providing insights into the IDSs scheme that would be able to detect them.

### Injection attacks

The injection attack sends packets with malicious data to the network. While the attacker injects malicious messages, the original sender still sends normal messages. From the perspective of the receiver node, the number of received messages is increased. Most of the attacks studied in IVN IDS schemes fall into this category, which can be subdivided into the following:

**DoS attack:** A DoS attack aims to exhaust the target ECU resources or deplete network bandwidth by sending massive legitimate requests. In the CAN bus, for instance, the adversary utilizes the priority arbitration mechanism to send a large number of high-priority messages (CAN ID:0x000), preventing other ECUs from transmitting their messages (*Studnia et al., 2013*; *Palanca et al., 2017*). The probability and catastrophic of a DoS attack are considered very high owing to adversaries with limited prior knowledge can implement this.

**Replay attack:** The purpose of a replay attack is to record previously propagated messages in the network by eavesdropping and re-transmit them later. For instance, The CAN bus is vulnerable to replay attacks due to a lack of freshness mechanisms. An attacker could access the engine speed signal and performs the replay attack, causing other ECU uses outdated information and impacting vehicle functionalities (*Hoppe & Dittman, 2007*).

**Spoofing attack:** The attackers inject fake messages into the network with the purpose of overlaying signals sent by the original ECU and deceiving the receiver (*Larson, Nilsson & Jonsson, 2008*). For example, the attackers only modify the corresponding bits of the CAN message representing the RPM (revolution per minute) gauge (*Martinelli et al., 2017*). While modifying a signal with a specific function requires the attacker to have prior knowledge of the communication matrix or execute reverse engineering to the traffics. Usually, this type of modification attack injects a small number of messages, making it challenging to identify.

**Fuzzing attack:** A fuzzing attack is used to investigate the impacts of different packets on the ECUs by transmitting messages with random protocol headers and payload into the network at a high frequency (*Martinelli et al., 2017*; *Fowler et al., 2019*). Unlike DoS attacks, inject packets may conform to normal traffic characteristics, causing the receiver node to use the information in the malicious packets with unexpected results.

**Table 3  Attacks on in-vehicle networks.**

|  | Features | Results | Attack types |
|---|---|---|---|
| DoS | Lots of legitimate requests appear in a short time. | Disturb the expected functionality of the system, and cause other nodes to stop their transmission. | T.T. |
| Replay | The re-transmit packets payload conforms to the pre-defined signal range. | The receiving ECU uses outdated signals resulting in vehicle misoperation. | T.T. |
| Spoofing | It just injects a small number of messages with malicious signals. | The spoofing attack deceives the receiver ECU, causing it to be distracted. | T.T. |
| Fuzzing | The protocol headers and payload values are random. | Spoofed random packets are rapidly injected into the network, causing unexpected operations. | T.T. |
| Suspension | The number of packets may decrease because the attacker suspends some or all messages. | It will affect normal operations and functionalities of the target ECU, and other receivers that rely on the constantly updated data of the target ECU. | T.T. |
| Masquerade | Masquerade attacks do not change the amount of traffic in a period; however, the payload value is out of context. | The attacker will alter vehicle controls, threatening the safety of passengers and road users. | T.O. |

**Notes.**

Abbreviations: T.T., Timing Transparent; T.O., Timing Opaque.

### Suspension attack

A suspension attack aims to stop some or all message transmission of the target ECUs, thus affecting the functionality of other ECUs that rely on this continuously updated information (*Verma et al., 2020a*). For example, the adversary can disturb the CAN frame with a particular ID to generate an error frame (*e.g.*, stuff error), which violates the CAN protocol specification, causing the target node to enter an error state. Besides, executing this attack on SOME/IP can intercept and discard the information *via* a man-in-the-middle (*Zelle et al., 2021*).

### Masquerade attack

The attacker first suspends a specific message and then transmits the spoofed packets with the same transmit interval, message format, and payload value range (*Iehira, Inoue & Ishida, 2018*). Usually, masquerade attacks do not change the amount of traffic in a period, which is more challenging to identify than injection attacks. The IDS scheme needs to check the reasonableness of the packet payload to detect these sophisticated attacks.

Here, we use the concepts of Timing Transparent (T.T.) and Timing Opaque (T.O.) provided by *Verma et al. (2020a)*, aiming to map these attack categories to different deep learning-based intrusion detection techniques. A Timing Transparent attack means it changes the frequency of network traffic, specifically the packet interval decrease/increase. In comparison, the Timing Opaque attack is defined as it does not disrupt network traffic timing or distributions. For example, the Timing Transparent attack, such as DoS or fuzzing attack, is detectable by the scheme using the CAN ID sequences. On the other hand, the Timing Opaque attack needs a more sophisticated scheme by checking payload data. In Table 3, we itemize the features, results, and types (T.T. or T.O.) of the discussed IVN attacks for comparison and reference.

| Timestamp | No. | Data | | | | | | | |
|---|---|---|---|---|---|---|---|---|---|
| | | D1 | D2 | D3 | D4 | D5 | D6 | D7 | D8 |
| 0.010 | A | 0.31 | 0.88 | 1 | 0 | 0 | 0.06 | 0 | 0 |
| 0.012 | B | 1 | 0 | 0 | 0 | 0 | 0 | 0 | 0 |
| 0.020 | A | 0.32 | 0.84 | 0 | 0 | 0 | 0.06 | 0 | 0 |
| 0.023 | C | 0.01 | 0.01 | 0.35 | 0.05 | 0.10 | 0.95 | 0.47 | 0.14 |
| 0.030 | A | 0.33 | 0.79 | 0 | 0 | 0 | 0.07 | 0 | 0 |
| ⋮ | ⋮ | ⋮ | ⋮ | ⋮ | ⋮ | ⋮ | ⋮ | ⋮ | ⋮ |

(a)

| Timestamp | No. | Signals of A | | | | Signals of B | | | Signals of C |
|---|---|---|---|---|---|---|---|---|---|
| 0.010 | A | 1 | 23.04 | 17.79 | 0 | - | - | - | - |
| 0.012 | B | - | - | - | - | 65.33 | 14.56 | 7 | - |
| 0.020 | A | 1 | 24.71 | 16.35 | 0 | - | - | - | - |
| 0.023 | C | - | - | - | - | - | - | - | 35.34 |
| 0.030 | A | 0 | 24.35 | 16.03 | 0 | - | - | - | - |
| ⋮ | ⋮ | ⋮ | ⋮ | ⋮ | ⋮ | ⋮ | ⋮ | ⋮ | ⋮ |

(b)

**Figure 4  Representation of the datasets.** (A) Raw data form. (B) Signal form.

## Datasets

Benchmark datasets are essential for training and evaluating the proposed DL-based IDS scheme. *Verma et al. (2020a)* have discussed the primary datasets used in the IVN IDS schemes and performed the quality analysis of both data and documentation. In this subsection, we detail a supplement about the characteristics in Table 4, like covered attack types and representations, to help researchers decide which datasets are most appropriate to their research context. We observed that different Ethernet application layer protocols have only one corresponding dataset compared to the CAN bus. And these existing datasets are specific to testing the proposed IDS, and their applicability is limited. Furthermore, the majority of datasets are limited to injecting simple messages and creating attacks by synthesizing data. Therefore, a point to be researched is that the scheme trained on a synthetic dataset needs to be evaluated in real environments. Besides, studied datasets are published in the form of raw data or signal sequences, as shown in Fig. 4. Some researchers (*Sun et al., 2021*) try to translate the raw data into signals and extract the signal boundary values by reverse engineering without prior knowledge of the communication matrix. However, *Marchetti & Stabili (2019)* indicated that the translation result is not completely correct.

# DEEP LEARNING-BASED IDS

In this section, deep learning methods used in IDS for in-vehicle networks are discussed. We detail the role of the adopted deep learning techniques in different stages of the intrusion detection procedure. In addition, the evaluation criteria and achieved results are presented. Our objective was to conduct a comprehensive survey of the development strategies for state-of-the-art IDS solutions that help researchers in deciding which network architecture is most suitable for their respective studies.

## DL-based IDS framework

This section proposes a generic deep learning framework for designing IDS in the in-vehicle networks domain. It is built upon the knowledge distilled from the articles investigated in this survey, as illustrated in Fig. 5. The first step in building an intrusion detection system is to consider the threat model and attack that are anticipated to be defended. In this step, the authors should clearly define the objective of the detector (*e.g.*, binary/multiclass classification). If a multiclass classification is required, the infected traffic must be further

Luo et al. (2023), *PeerJ Comput. Sci.*, DOI 10.7717/peerj-cs.1648

**Table 4 Open IDS datasets.**

| Dataset | Organization | Protocol | Real/Synthetic | DoS | Replay | Spoofing | Fuzzing | Suspension | Masquerade | Number of messages | Labeled |
|---|---|---|---|---|---|---|---|---|---|---|---|
| OTIDS (*Jeong & Kim, 2017*) | HCRL | CAN | Real | √ | – | √ | √ | – | – | 4,613 k | No |
| Car Hacking Dataset (*Kim, 2018*) | HCRL | CAN | Real | √ | – | √ | √ | – | – | 16,569 k | Yes |
| Survival Analysis Dataset (*Han & Kim, 2018*) | HCRL | CAN | Real | √ | – | √ | √ | – | – | 1,735 k | Yes |
| SynCAN (*Hansel-mann et al., 2019*) | Bosch | CAN | Synthetic | √ | √ | √ | √ | √ | – | 41,856 k | Yes |
| Automotive CAN Bus Intrusion Dataset v2 (*Dupont et al., 2019*) | TU Eindhoven | CAN | Real/Synthetic | √ | √ | √ | √ | √ | – | 3,176 k | No |
| CAN Signal Extraction and Translation Dataset (*Kim, 2020*) | HCRL | CAN | Real | – | – | – | – | – | – | 5,126 k | No |
| CAN Log Infector & Ambient CAN Traces (*Gazdag, 2020*) | CrySyS Lab | CAN | Real | – | – | – | – | – | – | 1,209 k | No |
| ROAD Dataset (*Verma et al., 2020b*) | ORNL | CAN | Real | – | – | √ | √ | – | √ | 28,244 k | Partial |
| Attack&Defense Challenge 2020 Dataset (*Kim, 2021a*) | HCRL | CAN | Real | √ | √ | √ | √ | – | – | 8,694 k | Yes |
| SOME/IP Dataset (*Alkhatib, Ghauch & Danger, 2021*) | LTCI | SOME/IP | Synthetic | – | – | √ | – | √ | – | 12k | Yes |
| Automotive Ethernet Intrusion Dataset (*Kim, 2021b*) | HCRL | AVTP | Real | – | √ | – | – | – | – | 1,941 k | No |

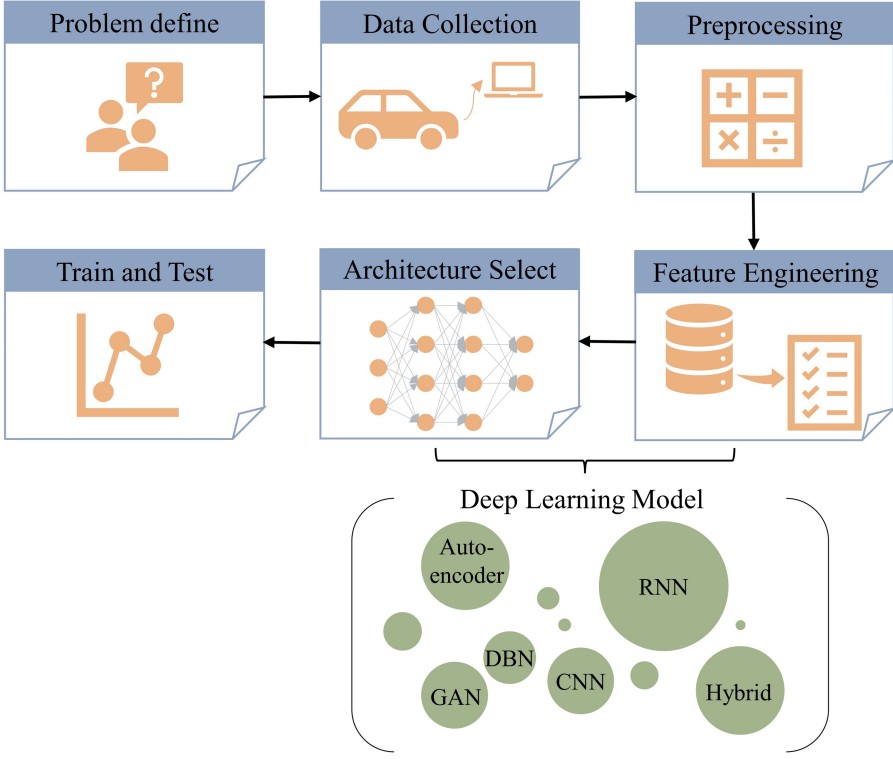

**Figure 5** **Deep learning-based IDS framework.**

classified into different classes and subclasses of attacks. Then, collecting a vast amount of data with sufficient quality is necessary. In practice, researchers often collect a dataset by monitoring the real vehicle communication or building a testbed composed of physical or virtual nodes. No dataset is perfect, as measurements often include artifacts that can impact the characteristics, such as different driving preferences and movement tracks.

After acquiring a dataset, its characteristics need to be carefully assessed to understand not only the content of the data but also its flaws. In the preprocessing step, authors need to clean, merge and convert the data into suitable formats and types. Extract and select appropriate features, for instance, raw data fields (CAN ID, data byte, ethernet destination Mac address, source Mac address) or statistical traffic characteristics (the transmission frequency of a specific message over some time). Subsequently, the authors select the appropriate deep learning model depending on the attack paradigm and extracted features. The modeling process is iterative, with repeated experiments to find the optimal hyperparameters set that makes the model perform optimally on the dataset. Finally, evaluating an intrusion detection system using multiple datasets can let it adapt to different environments.

## The proposed DL-based IDS approaches

This section comprehensively researches and comparatively analyzes a significant number of in-vehicle IDS approaches using deep learning technologies. Figure 6 presents a

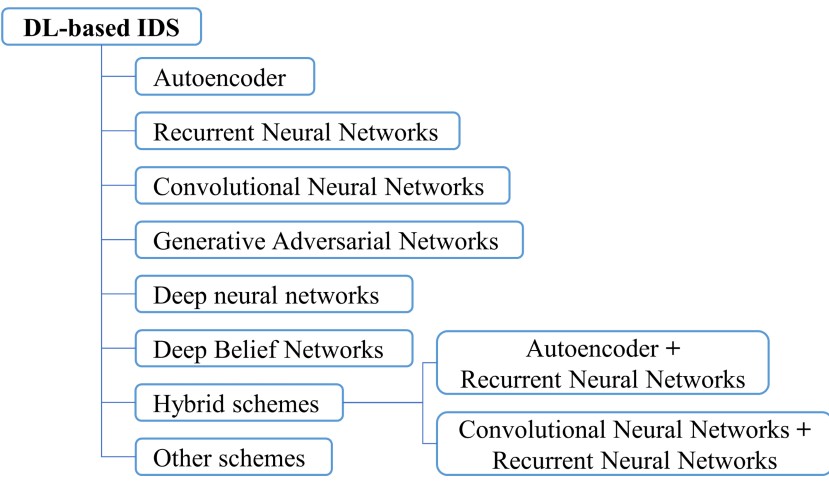

**Figure 6** Taxonomy of the deep learning-based IDS schemes.

taxonomy of the researched schemes according to the deep learning network architectures utilized, including based on autoencoder (AE) structure, recurrent neural networks (RNN) structure, convolutional neural network (CNN) structure, generative adversarial network (GAN) structure, deep neural networks (DNN) structure, and deep belief networks (DBN) structure. This section is organized based on this classification method. The corresponding subsection is assigned to describe the main contributions of different deep learning architectures used in IDS schemes, and their properties are compared in terms of targeted application protocol, feature selection, used datasets for model training and testing, evaluation metrics, classification type, and the employed deep learning methodology.

### Autoencoder

AE is the common DL technique applied in IDS research for dimensionality reduction and anomaly detection in an unsupervised way (*Hinton & Salakhutdinov, 2006*). It generally consists of input and output layers with the same dimensions, an encoding neural network, a decoding neural network, and a latent space. AE works on matching the decoder output as close to the encoder input as possible by performing representation learning, a process called reconstruction. Therefore, It is trained to minimize the reconstruction error between the input and the reconstruction sequence to obtain a low-dimensional abstraction representing high-dimensional data. Then, an input traffic sequence is considered abnormal if its reconstruction error is larger than the specified threshold. This subsection discusses the AE-based IDS schemes (*Alqahtani & Kumar, 2022*; *Cheng, Han & Liu, 2023*; *Alkhatib et al., 2022*) implemented for the in-vehicle network.

*Lin et al. (2020)* proposed an efficient DL-based IDS scheme using the deep denoising autoencoder, which can learn latent sequential patterns of CAN data. This scheme adds an evolutionary-based optimization algorithm to tune the proposed model parameters, aiming to maximize the performance and efficiency of the model against malicious attacks. This solution is evaluated on the three datasets, including two popular anomaly

detection datasets in the area of the CAN bus. They achieved high precision and F1-score, demonstrating that this model is robust and reliable in detecting DoS attacks, fuzzing attacks, and impersonation attacks.

*Wei et al. (2023)* introduced a NIDS approach, denoted as AMAEID, for CAN bus real-time intrusion detection using the denoising autoencoder and attention mechanism. The hexadecimal CAN payload was converted into binary format in the data preprocessing stage. Then it superimposed the noise conforming to the Gaussian distribution, which benefited the model in terms of interference resistance and generalization. Finally, a single-layer fully-connected network is used to obtain the final prediction value, indicating whether the message is normal or abnormal. The authors validated their model using OTIDS dataset and demonstrated that it achieves high performance. However, this scheme is only evaluated with traditional machine learning methods (*e.g.*, DT, KNN, SVM), and it is not analyzed with other literature using AE to justify the achieved results.

*Lokman et al. (2019)* proposed an IDS approach by applying deep contractive autoencoders (DCAEs) to extract flow-based features from network traffic data. This scheme imposes different penalty terms on the loss function representation, ensuring the network is robust to minor changes, noise, and missing inputs in the training samples. This scheme is evaluated using real CAN data collected from three brand vehicles through OBD-II. The results are compared with stacked sparsed autoencoders and denoising autoencoders.

The NIDS scheme proposed in *Ashraf et al. (2021)* utilized the long short-term memory (LSTM) autoencoder algorithm on dealing with attack events that occur at the central network gateways of vehicles. This scheme executes binary classification on incoming traffic. Statistical features of the mean and standard deviation computed for messages within a specific time window are transformed by the normalized likelihood sequence and fed into the network. The author validated their method using the UNSW-NB15 dataset for external network communication and the car hacking dataset for in-vehicle communication. The experimental results on two datasets demonstrate that the scheme achieves 99% and 98% accuracy, respectively.

*Longari et al. (2021)* also proposed an IDS method by applying LSTM autoencoders to extract latent features from malicious behaviors. The proposed model creates a time series of CAN payload for each CAN ID and learns to update network hyperparameters to minimize the Mahalanobis distance between the reconstructed sequence and the original sequence. Larger reconstruction errors will be flagged as potential exceptions. The model is evaluated using collected CAN frames during vehicles driving in different conditions, such as cities and highways. The authors indicate that the proposed method can effectively perform binary classification of intrusion data by performing the required experiments. However, CANnolo has conspicuous detection time consumption and is unsuitable for strong real-time systems.

*Kukkala, Thiruloga & Pasricha (2020)* proposed a gated recurrent unit (GRU) autoencoder network for detecting intrusive and anomalous flow-based data. Compared to LSTM-based autoencoders, the training process of this scheme consumes less time. The author designed this IDS approach focusing on monitoring the CAN message signal, so

the method is protocol agnostic and not limited to the CAN bus protocol. In addition, the authors propose an intrusion score metric to measure the degree of deviation from normal system behavior to determine whether a signal is normal or abnormal. They evaluated their model using the synCAN dataset published by Bosch and analyzed the memory footprint, inference time, and scalability.

*Hoang & Kim (2022)* introduced a semi-supervised learning-based IDS approach by incorporating convolutional autoencoder and GAN. This scheme trains a classifier in three separate stages. In its first stage, it uses unlabeled data to train the autoencoder part. In its next phase, the encoder and discriminator are trained together. Then update the encoder again with the labeled data by minimizing the cross-entropy loss. This IDS scheme is tested using public datasets. Experimental results demonstrate that feeding only 40% of labeled data to the model achieves the lowest error rate of 0.1% compared to other supervised methods. The properties of AE-based IDS solutions described in this subsection are compared in Table 5.

### RNN

Recurrent neural networks (RNNs) extend the capacities of the traditional feed-forward neural network with its ability to learn sequential data over timesteps, especially suitable for sequences that are not independent of each other or involve contextual associations. It consists of the input, hidden, and output units, where the output of the hidden unit relies on its input at the current time step and the output of the earlier time step. The network parameters are updated through the back-propagation over time (BPTT) algorithm. RNNs may provide a good idea for intrusion detection by predicting the upcoming signals of the near future received frames. The detector then regarded the unexpected deviation from the actual signal as an intrusion. However, RNNs usually only can handle finite length sequences and will encounter the vanishing gradient problem if the output at any given timestep depends on inputs much earlier. Different RNN variants like LSTM and GRU are presented to address these issues.

Several RNN-based in-vehicle intrusion detection approaches, such as (*Ma et al., 2022*; *Tanksale, 2020*; *Tanksale, 2021*; *Desta et al., 2020a*; *Desta et al., 2020b*; *Tariq et al., 2019*; *Mansourian et al., 2023*; *Al-Jarrah et al., 2023*), are proposed in the literature, which will be discussed in this section. RNN-based IDS was first applied by *Taylor, Leblanc & Japkowicz (2016)* to detect malicious behavior on the CAN bus. The classifier works by learning to predict the next frame according to the message previously sent on the bus. The authors validated their anomaly detection method using the abnormal data synthesized by adding CAN frames, deleting expected frames, and modifying data contents. Experimental results show that this scheme can successfully flag suspicious anomalies. Nevertheless, a separate detector was trained for each CAN ID without considering the dependencies between different IDs.

*Qin, Yan & Ji (2021)* presented a NIDS established on the LSTM technique, identifying abnormals and keeping a long-term memory of them. This research focuses on time series prediction, calculating the loss between the predicted content and that transmitted on the bus at the next time step. Then, it is flagged whether the packet is abnormal or normal based

Luo et al. (2023), *PeerJ Comput. Sci.*, DOI 10.7717/peerj-cs.1648

**Table 5** Properties of AE-based IDS schemes.

| Scheme | Protocol | Input features | Dataset | Effectiiveness (%) | | B/M[*] | Detect time | Methodology |
|---|---|---|---|---|---|---|---|---|
| | | | | Accuracy | F1 | | | |
| *Lin et al. (2020)* | CAN | CAN ID | OTIDS & Self-Collected | – | 98 | B | – | Denoising Autoencoder with ecogeography-based optimization algorithm |
| *Wei et al. (2023)* | CAN | Payload | OTIDS | – | – | B | – | a denoising autoencoder with the attention mechanism |
| *Lokman et al. (2019)* | CAN | CAN ID & Payload | Self-Collected | – | – | B | – | deep contractive autoencoder |
| *Ashraf et al. (2021)* | CAN | statistical features | car-hacking dataset UNSW NB-15 | 99 96 | 99 98 | B | – | A Deep Learning architecture-based LSTM autoencoder |
| *Longari et al. (2021)* | CAN | Payload | Self-Collected | – | 94.83 | B | 3.5 GHz single core CPU: 950ms | a LSTM-based recurrent autoencoder |
| *Kukkala, Thiruloga & Pasricha (2020)* | CAN | Payload | SynCAN | 99 | – | B | Jetson TX2 board with an ARM Cortex-A57 CPU: 0.08ms | a GRU-based recurrent autoencoder |
| *Hoang & Kim (2022)* | CAN | CAN ID | car-hacking dataset | – | 99 | B | Intel Core i7-7700 CPU 3.6 GHz and a GTX1060 GPU: 0.63ms | a semi-supervised learning-based convolutional adversarial autoencoder model |

**Notes.**
  *"B" and "M" refer to binary-classification and multi-classification, respectively.

on the deviation of the loss value from the threshold. They evaluated the performance of the proposed model on multi-format data input and five loss functions. Nevertheless, this method is not tested using other publicly available datasets.

*Zhu et al. (2019)* proposed a mobile edge-assisted anomaly detection method for IVNs based on LSTM. It alleviates the problem of insufficient computing resources for onboard devices when deploying deep learning-based anomaly detection systems to vehicles. The model consists of a local common hidden layer computation unit and two single LSTM neural networks that separately process time-based and data-based features. The necessary experiments indicated that with the assistance of the mobile edge, their IDS approach reaches 0.9 detection accuracy, and a CAN message detection is completed in an average of 0.61 ms. However, applying the MEC server introduces indeterminate network latency when delivering CAN bus messages to the MEC server.

*Alkhatib, Ghauch & Danger (2021)* present an RNN-based IDS to handle the offline intrusion detection problem on SOME/IP application layer protocol. They deal with class imbalance using the adaptive weighting technique that weights samples in rare classes at high costs. The experiments are conducted on the datasets constructed using a packets generator implemented in python language. Results validate that the proposed RNN-IDS provides good results with F1 Scores and AUC values of each intrusion type. However, this scheme merely evaluates attacks that deviate from the protocol specification in the inter-device communication session. In another work (*Luo et al., 2023*), the authors conducted a multi-layered IDS with GRU-based modules that could simultaneously detect anomalies on SOME/IP header and payload.An LSTM-based IDS was presented by *Hossain et al. (2020)* in the binary and multi-class classification of the CAN bus DoS attacks, fuzzing attacks, and spoofing attacks. Several layer LSTM networks are utilized to identify cybersecurity attacks, and their outputs are fed into a softmax function. They also investigated the impact of different learning rates, activation functions, loss functions, the number of neurons, and optimizers on model performance with Vanilla LSTM and Stacked LSTM, respectively. This model is evaluated on the self-collected dataset and Survival Analysis datasets, and the results indicate the effectiveness of their approach by comparing it with the Survival Analysis method.

*Tariq, Lee & Woo (2020)* proposed CAN-ADF, a hybrid DL-based IDS model for detecting anomalies and intrusions formulating as a multiclass classification problem. More specifically, this method uses heuristics-based and LSTM-based methods to identify attacks in in-vehicle network traffic. This combination is because diverse anomalies and attacks arise with varying levels of detection difficulty. Methods with different algorithms may be good at detecting different types of attacks. They evaluate the detection performance of the proposed model by collecting 7,875,791 CAN messages from KIA Soul and Hyundai Sonata. The experimental results demonstrate that the proposed scheme has an average accuracy of 99.45% on multi-classification problems.

*Khan et al. (2021)* introduced a multi-stage intrusion detection method by combining a normal state-based bloom filter and bidirectional long short term memory (BiLSTM) to discover attacks from communication networks of AVs efficiently. They indicated that bloom-filter with the advantages of efficient memory usage, light-weighted, and constant

lookup time to recognize whether network data matches the normal communication database before being fed into the neural network. The authors evaluated their intrusion detection model. The results show that the model achieves an accuracy of 99.11% on the car-hacking dataset and 98.88% on the UNSW NB-15 dataset, representing CAN bus traffic and IP-based network traffic, respectively.

*Driss et al (2022)* proposed a cyberattack detection framework for vehicular sensor networks that applies GRU. This IDS scheme considers the relationship between previous and present data, which can improve detection capability and rate. The authors employ a group of GRU with a Random Forest (RF)-based ensemble unit to solve the problem of privacy data and resource limitation of intelligent sensor networks. They tested their anomaly detection approach and applied metrics such as accuracy, recall, precision, and F1 score. Experimental results also indicated that their approach outperformed other classifiers. Table 6 presents the various aspects of the RNN-based IDS approaches.

### CNN

Convolutional neural networks (CNNs) are specialized neural networks destined to perform image processing and analysis. It generally consists of a stack of input, convolutional, pooling, and fully connected layers. Compared with other deep learning networks under the same network depth, CNN requires fewer parameters, reduces complexity, and speeds up the learning process since CNN adopts local connections and weight sharing instead of traditional fully connected networks. Indeed, It can be usually applied in traffic intrusion detection to perform supervised feature extraction and classification tasks by organizing network traffic into multiple arrays, such as the 1D array of CAN IDs and signals.

This section investigates the CNN-based IDS schemes presented in the literature, such as *Lin et al. (2022)*, *Ahmed, Jeon & Ahmad (2023)* and *Taslimasa et al. (2023)*. *Song, Woo & Kim (2020)* presented an IDS scheme, which preprocesses the CAN bus traffic data and then models the temporal sequential patterns using deep convolutional neural networks. This scheme directly converted the bit-stream data of the CAN bus into a grid-like structure, more specifically, building the 29-bit IDs of consecutive 29 frames into a 29*29 matrix as features of the CNN. The IDS scheme adopted the Inception-ResNet, which performs well in natural image classification tasks, to detect abnormal data by reducing the original model components and tuning the network hyperparameters based on the difference in feature dimensions. By conducting the experiments, the authors indicated that this model achieves better detection results compared with LSTM, ANN, and as well as SVM, KNN, NB, and decision tree. In particular, the IDS shows better performance when detecting fuzzy attacks. Moreover, the authors conducted offline testing on devices with GPUs and only CPUs. The results exhibited that the proposed model is challenging to employ online detection in existing automobiles by calculating the detection latency of the processing packets.

*Desta et al. (2022)* proposed Rec-CNN, a CNN-based IDS technique for investigating network flow data for abnormal activities. This method applies the categorical recurrence plots algorithm, which creates images from time-series data on the CAN bus. Unlike constructing images in *Song, Woo & Kim (2020)*, the authors utilize recurrence plots to

**Table 6  Properties of RNN-based IDS schemes.**

| Scheme | Protocol | Input Features | Dataset | Effectiiveness (%) | | B/M* | Detect time | Methodology |
|---|---|---|---|---|---|---|---|---|
| | | | | Accuracy | F1 | | | |
| *Taylor, Leblanc & Japkowicz (2016)* | CAN | Payload | Self-collected | – | – | B | – | a LSTM-based neural network to predict the next data word |
| *Qin, Yan & Ji (2021)* | CAN | Payload | Self-Collected | – | 85 | B | – | a LSTM-based neural network to predict the next frame payload |
| *Zhu et al. (2019)* | CAN | Payload | Self-Collected | 89.3 | 90.2 | B | 3.3 GHz dual core Intel i5 CPU: 0.61ms | a multi-dimension LSTM framework to predict information of next CAN message |
| *Alkhatib, Ghauch & Danger (2021)* | SOME/IP | Protocol header | Self-Collected | – | 80 | B | – | a two layers RNN and a dense layer with softmax |
| *Luo et al. (2023)* | SOME/IP | Protocol header & Playload | Self-Collected | 99.77 | 99.6 | B | Jetson Xavier NX: 0.36ms | combination of rule-based and GRU based methods |
| *Hossain et al. (2020)* | CAN | CAN ID & Playload | Self-Collected; Survival Analysis Dataset | 99.99 99.99 | 99 99 | B & M | – | several layers LSTM and a dense layer with softmax |
| *Tariq, Lee & Woo (2020)* | CAN | CAN ID & Playload | Self-Collected | 99.54 | 99 | M | Intel Xeon E5-1650 CPU and GTX 1080ti GPU: 73ms | combination of rule-based and LSTM based methods |
| *Khan et al. (2021)* | CAN | PCA | car-hacking dataset; UNSW NB-15 | 99.11; 98.88 | 99.09; 98.85 | B | Intel i5 3.20 GHz: 0.023 ms | a multi-stage intrusion detection framework based Bi-LSTM and bloom-filter |
| *Driss et al (2022)* | CAN | – | Car Hacking: Attack & Defense Challenge 2020 Dataset | 99.52 | 98.92 | B | – | a group of GRU with a Random Forest (RF)-based ensembler unit |

**Notes.**

*"B" and "M" refer to binary-classification and multi-classification, respectively.

capture the temporal dependencies in the sequence of arbitration IDs. They compare the proposed scheme with *Song, Woo & Kim (2020)*, and the experimental results indicate that it achieves better detection performance with comparable execution time.

*Jeong et al. (2020)* designed a CNN-based IDS to identify the legitimacy of communication nodes, analyzing the channel characteristics between a specific pair of transmitters and receivers. This model takes the bus differential voltage signal as input, outputs the probability that the received message is transmitted from each node, then identifies the node with the highest value as the predicted message source. Then, the detector compares the predicted source with the actual source node number obtained from the mapping table, and the inconsistency is considered abnormal. Simultaneously, the detector continuously transmits dominant bits to generate error frames to interrupt the transmission of abnormal messages. However, if the detector occurs a false alarm, it may seriously block the transmission of normal messages and reduce the CAN bus network performance.

*Jeong et al. (2021)* introduced a NIDS using CNN to detect AVTP packet replay attacks in automotive Ethernet-based networks. This scheme extracts features from the first 58 bytes of the AVTP stream, inferring the stream AVTPDU is benign or not. The authors captured real AVTP packets as training and testing datasets by building a physical testbed based on BroadR-Reach. Furthermore, they also measure the detection latency on hardware devices such as Google collab, macintosh, Jetson TX2, and Raspberry Pi 3. Results validate that this scheme is efficacious for binary classifications of abnormal traffic.

*Han, Kwak & Kim (2023)* presented the TOW-IDS solution for heterogeneous in-vehicle networks, which detects the abnormalities of automotive ethernet. This scheme constructed a DCNN model by adjusting the ResNet model as the classification algorithm and furthermore incorporated wavelet transform to reduce the image data size. In consideration of the length difference in different protocol packets (such as CAN, gPTP, and AVB), the authors padded and trimmed the packets to satisfy the computation condition during the data preprocessing process. They evaluated the performance of their approach in identifying frame injection, PTP sync, MAC Flooding, CAN DoS, and CAN replay attacks on self-collected datasets. Experimental results demonstrated that the TOW-IDS consumes less time in detecting network anomalies than default ResNet and EfficientNet methods. In Table 7, the features of discussed CNN-based IDS schemes are listed.

### GAN

GAN is a deep generative model developed by *Mirza & Osindero (2014)*, which consists of two ANN networks, namely a generative network and a discriminate network. The generative network learns the distributions of the real data and attempts to generate samples with the same characteristics to confuse the discriminator, which in turn endeavors to distinguish the real data from the generated. Researchers usually utilize GAN models to alleviate the problem of insufficient attack samples in intrusion detection.

*Seo, Song & Kim (2018)* presented a GAN-based IDS for in-vehicle networks to identify normal and attack traffic for the first time. The generator of the GAN model is composed of CNN, and the discriminator is comprised of DNN. They grouped CAN IDs in the order of

Luo et al. (2023), *PeerJ Comput. Sci.*, DOI 10.7717/peerj-cs.1648

**Table 7   Properties of CNN-based IDS schemes.**

| Scheme | Protocol | Input features | Dataset | Effectiiveness (%) | | B/M[*] | Detect Time | Methodology |
|---|---|---|---|---|---|---|---|---|
| | | | | Accuracy | F1 | | | |
| *Song, Woo & Kim (2020)* | CAN | CAN ID | car-hacking dataset | – | 99.9 | B | two 2.30 GHz Intel Xeon CPUs and a Nvidia Tesla K80 GPU. CPUs only: 6.7 ms: With GPU acceleration: 5 ms | a DCNN model constructed by Reducing Inception-ResNet components |
| *Desta et al. (2022)* | CAN | CAN ID | car-hacking dataset: Self-Collected | 99.9 | – | B & M | Jetson TX2: 117ms | a simple two-layered CNN that uses recurrence plots to generate images as input |
| *Jeong et al. (2020)* | CAN | physical characteristics | Self-Collected | 99.92 | – | B | – | The CNN is composed of an input layer, a fully connected (FC) layer, a softmax layer, and two hidden sub-layers. |
| *Jeong et al. (2021)* | AVTP | Protocol header | automotive-ethernet-intrusion-dataset | 99.55 | 99.27 | B | Jetson TX2: 0.982 ms Raspberry Pi 3 with ARM Cortex-A53 CPU: 35 ms | CNN model comprises an input layer, two hidden sub-layers, and two dense layers. |
| *Han, Kwak & Kim (2023)* | CAN & AVTP & gPTP | Protocol header & Payload | Self-Collected | 99.65 | 99.7 | B | 4,790K CPU and 2080 RTX GPU: 29.1 ms | A DCNN model formed by adjusting the ResNet algorithm, and cooperates with wavelet transform to reduce the image data size. |

**Notes.**

[*] "B" and "M" refer to binary-classification and multi-classification, respectively.

sending time and converted them into sample images after one-hot encoding. The authors evaluated their scheme for detecting DoS attacks, fuzzy attacks, and spoofing attacks with the car hacking dataset.

*Yang et al. (2021)* attempted to train the proposed IDS scheme using the generated CAN images constructed with frames interval time, CAN ID, and DLC. Due to employing a different encoding algorithm compared with (*Seo, Song & Kim, 2018*), this scheme reduces the input data dimension and the number of network layers and neurons. Furthermore, this scheme applies a sparse enhancement training method benefiting the discriminator to correct its learning direction and accelerate the training convergence, thereby overcoming the poor performance of single GAN. The authors compared the achieved results against (*Seo, Song & Kim, 2018*) using metrics such as accuracy, recall, precision, and F1 score. The evaluation results outperform related works in terms of precision and accuracy.

*Xie et al. (2021)* extend the ability of the discriminator to detect tampering attacks by introducing CAN communication matrix that indicates the signal maximum and minimum range. In the data preprocessing stage, the CAN messages are divided into five types according to the transmission mode, effectively reducing discriminator misjudgment. In addition, Considering that the frames sequence is not fixed in the driving environment, the author uses 64 consecutive messages of the same ID to construct the CAN image instead of directly using traffic data that may include different CAN IDs. Experimental results illustrate that their IDS approach could enhance detection performance.

*Zhao et al. (2022)* proposed an IDS model for implementing multi-classification of CAN bus frames using the auxiliary classifier generative adversarial network (ACGAN). They propose and evaluate the performance of four different variants and indicate that the two-stage classification architecture of ACGAN combined with out-of-distribution works best. ACGAN assigns fine-grained labels to known attacks in the first stage. Out-of-distribution samples are passed into the binary real-fake classifier to perform unknown attack classification. This scheme is tested on public datasets, and the results indicate that the proposed scheme reaches a higher classification accuracy with lower resource overhead. It is suitable for employment in resource-constrained in-vehicle environments. As shown in Table 8, some effective GAN-based IDS models are compared.

### DNN

DNNs can be considered an ANN structure with additional depth. It increases the number of hidden layers, enhancing the abstraction capabilities of the model to learn complex nonlinear relationships. DNNs are typical feed-forward networks where data flows unidirectionally from the input layer to the output layer.

This section investigates the articles that utilized DNN techniques to detect in-vehicle network intrusions. *Zhang et al. (2019)* proposed an in-vehicle IDS based on DNN approach, which attempts to detect spoofing and replay attacks by extracting vehicle behavior features from communication traffics. The author adopts two enhanced BP algorithms, gradient descent with momentum (GDM) and gradient descent with momentum adaptive gain (GDM/AG), to speed up the iterative update process of the network. They collected nearly 300,000 CAN bus messages using the open-source software

**Table 8  Properties of GAN-based IDS schemes.**

| Scheme | Protocol | Input Features | Dataset | Effectiiveness (%) | | B/M[*] | Detect time | Methodology |
|---|---|---|---|---|---|---|---|---|
| | | | | Accuracy | F1 | | | |
| *Seo, Song & Kim (2018)* | CAN | CAN ID | car-hacking dataset | 98 | – | B | – | The generator is composed of CNN and the discriminator is composed of DNN |
| *Yang et al. (2021)* | CAN | CAN ID | car-hacking dataset | 99.8 | 99.8 | M | Intel Xeon CPU E5-2673 v3 @ 2.40 GHz and an NVIDIA GP102 GPU: 0.12 ms | GAN model uses CNN |
| *Xie et al. (2021)* | CAN | CAN ID & Payload | car-hacking dataset | – | 99.8 | M | a Xilinx Spartan 6 FPGA: 0.09 ms | GAN model |
| *Zhao et al. (2022)* | CAN | CAN ID | car-hacking dataset | – | 99.23 | M | Raspberry Pi 4 Model B and quad-core ARM Cortex-A72 CPU at 1.5 GHz. single-core: 0.538 ms multi-core: 0.203 ms | Auxiliary Classifier GAN |

**Notes.**
*"B" and "M" refer to binary-classification and multi-classification, respectively.

BusMaster and evaluated this scheme in terms of accuracy, TPR, and FPR. Experiments indicate that the proposed model improves a great deal about the internal cybersecurity of the onboard system.

*Cuzzocrea, Mercaldo & Martinelli (2020)* employed an MLP-based classifier to recognize DoS attacks, fuzzy attacks, and spoofing attacks on the CAN bus. The proposed method is assessed through experiments on the car-hacking dataset, where data field features are selected. After performing classification experiments using MLPs with multiple hidden layers, they indicated that the MLP network with 1 and 3 hidden layers obtained the best results.

*Zhang & Ma (2022)* presented a NIDS benefiting from both ruled-based and DNN-based approaches. This ensemble uses deep learning techniques to achieve high detection accuracy while reducing the computational requirements through offloading traffic with a rule-based module. Before conducting detection using the DNN classifier with five hidden layers, the model uses the pre-defined rules to catch malicious messages such as invalid ID, time intervals, and DLC. The authors selected five features with high importance scores in feature engineering for training instead of using the raw data directly. They tested their system on the datasets originating from four vehicles CAN bus data and achieved a binary classification accuracy of 99.8%. However, the model is not evaluated by employing other public datasets to verify its claimed results further. The characteristics of the DNN-based IDS schemes studied in this part are given in Table 9.

### DBN

DBNs are probabilistic generative models by stacking several restricted Boltzmann machines (RBM) in layers. In DBN, The network connection of the upper two layers is undirected, and the other layers are directed. DBN is pre-trained using an unsupervised greedy layer-wise learning method followed by learning valuable features using the supervised fine-tuning approach. For IDS, DBNs are usually used to initialize network parameters, improving anomaly detection accuracy. Besides, DBNs are applied to solve ANNs training problems, such as slow training and needing extensive labeled data.

The IDS schemes of this subsection incorporate DBN for in-vehicle communication abnormal classifying. *Kang & Kang (2016b)* presented an IDS method based on a deep belief neural network. First, they applied DBN to pre-train the neural network parameters intending to initialize the model, followed by the parameters tuned through supervised learning to obtain a better classification performance. The authors utilized all the DATA field for generating the feature. This approach has low computation complexity in the decision, but the training phase is time-consuming. In another article *Kang & Kang (2016a)*, the authors trained and evaluated the proposed DBN-based approach using simulated vehicular network communication called Open Car Testbed and Network Experiments (OCTANE) and simulated the attacker tampering with the tire pressure monitoring signal. They demonstrated that the detection accuracy of this approach is enhanced compared with feed-forward ANN. The characteristics of these two IDS schemes are shown in Table 10.

Luo et al. (2023), *PeerJ Comput. Sci.*, DOI 10.7717/peerj-cs.1648

**Table 9  Properties of DNN-based IDS schemes.**

| Scheme | Protocol | Input Features | Dataset | Effectiiveness (%) | | B/M[*] | Detect time | Methodology |
|---|---|---|---|---|---|---|---|---|
| | | | | Accuracy | F1 | | | |
| *Zhang et al. (2019)* | CAN | – | Self-Collected | 98 | – | B | inter CoreTM i5 CPU 1.80 GHz: 4 ms | DNN uses GDM and GDM/AG |
| *Cuzzocrea, Mercaldo & Martinelli (2020)* | CAN | Payload | car-hacking dataset | – | 98 | B | – | MLP classification with one/three hidden layers |
| *Zhang & Ma (2022)* | CAN | CAN ID & Payload | Self-Collected | 99.8 | – | B | Intel Core i5-4200U CPU 1.60 GHz: 0.55 ms | DNN-based IDS combines traditional rule-based techniques |

**Notes.**
*"B" and "M" refer to binary-classification and multi-classification, respectively.

**Table 10  Properties of DBN-based IDS schemes.**

| Scheme | Protocol | Input features | Dataset | Effectiiveness (%) | | B/M[*] | Detect time | Methodology |
|---|---|---|---|---|---|---|---|---|
| | | | | Accuracy | F1 | | | |
| *Kang & Kang (2016b)* | CAN | Payload | Self-Collected | 97.8 | – | B | 3.4 GHz intel CPU: 8.12 ms | DBN |
| *Kang & Kang (2016a)* | CAN | Payload | Self-Collected | – | – | B | – | RBM |

**Notes.**
[*]"B" and "M" refer to binary-classification and multi-classification, respectively.

### Hybrid

This section discusses several IDS schemes that combine two or more deep learning algorithms in detecting intrusions procedure.

*AE+RNN.* *Li et al. (2020)* have incorporated vanilla RNN and AE for in-vehicle anomaly and intrusion detection. A sparse autoencoder with L1 regular term constraints to the loss function is applied to extract network data in-depth features and reduce data dimensionality, followed by a classifier constructed by RNN and softmax for binary classification. The authors collected data using a USB-to-CAN converter directly connected to the OBD-II and conducted the required experiment. They indicated that the average processing time of this IDS scheme is significantly reduced compared to other deep learning-based studies. However, note that the literature compared is not newly published. In *Nichelini et al. (2023)*, authors present a modular framework that also includes the RNN autoencoder-based module.

*Hanselmann et al. (2020)* introduced CANet, a hybrid IDS scheme benefiting from the LSTM and auto-encoder, aiming to consider temporal dynamics and interdependencies between IDs. They employed a separated LSTM input model for each ID, then aggregated all input models and fed them into an autoencoder structure. Then, It applies the reconstruction error of the auto-encoder as the anomaly score for classification. Accordingly, the model is able to detect messages with different IDs simultaneously. The authors evaluate performance on both real and synthetic CAN data. Results revealed that this model performs better than approaches using only LSTM and auto-encoder.

*CNN+RNN.* A few schemes have incorporated CNNs and RNNs to improve anomaly and intrusion detection performance, such as (*Aldhyani & Alkahtani, 2022*; *Cherdo et al., 2023*). *Lo et al. (2022)* presented HyDL-IDS, which utilizes CNNs to learn the spatial–temporal features among network traffic and employs LSTMs to learn the temporal features further. This scheme extracts spatial–temporal features in characterizing network traffic more accurately than manual feature engineering. The authors evaluated this model using the car hacking dataset collected by the HCRL laboratory. The experimental results demonstrate that it achieves nearly 100% detection accuracy, and the false alarm rate is reduced compared to other methods such as Decision tree, CNN, and LSTM.

Also, *Agrawal et al. (2022)* designed an unsupervised learning-based IDS that exploits CNNs and LSTMs to detect anomalies in CAN networks. They use the sliding window approach to construct CAN traffic sequences in the data preprocessing stage. Each sequence

is divided into a normal or an attack type based on the percentage of abnormal data it contains. The normal type sequences are reconstructed during model weight training through a 1D convolution, LSTM units, and dense layers. Furthermore, the authors use the confidence interval-based approach to estimate a reconstruction error threshold, which could reduce the training time and manual work. Time-consuming experiments on the Nvidia Jetson Nano development board show that the scheme can be applied to embedded devices.

*Song & Kim (2021)* presented a self-supervised method employing noised pseudo normal data, which enables the model to detect unknown attacks on in-vehicle networks. The LSTM-based generator generates the noised pseudo normal data and uses them to train the CNN-based detector with normal data together. Unlike GAN-based intrusion detection methods, there is no minimax game in the training process of the generator and detector of this model. In addition, this method can alleviate the dataset imbalance problem caused by insufficient attack data. The evaluation results show that the model performs well in detecting known attacks and improves the detection ability against unknown attacks.

*Sun et al. (2021)* presented an IDS scheme to detect CAN frame anomalies using CNN and Bi-LSTM with the attention mechanism. This scheme uses 1D convolution layers to capture the local features of continuous physical signals and employs the Bi-LSTM to learn temporal features between multiple frames. Besides, it uses the attention mechanism to calculate the weights at different time steps for quickly focusing on network traffic key features. The authors implemented the proposed method using the Keras and indicated that the detection time achieved 5.7 ms on the real vehicle. However, the CLAM trains a set of network parameters for each CAN ID, which requires more computing and memory capacity for the device.

*Javed et al. (2021)* introduced CANintelliIDS, an IDS approach that integrates CNNs and attention-based GRU for detecting both single and hybrid attacks. The mixed attacks mean that attackers use different combinations of intrusion methods such as fuzzy, DoS, and impersonation, which is challenging for IDS. They performed the needed experiments using the OTIDS dataset and evaluated the accuracy and F1 score when it was imbalanced. The results show that this scheme effectively captures the latent characteristics of historical traffic compared to simply using the CNNs network. Table 11 provides a comparison of hybrid DL-based IDS models investigated in this subsection.

### Other

This section describes some other schemes for detecting intrusion attacks on the IVNs.

**Temporal convolutional network (TCN):** *Shi et al. (2021)* presented an anomaly detection system based on the TCN for detecting in-vehicle network attacks. The scheme predicts the sequences by extracting latent features of normal data, flags intrusion when an unknown sequence occurs and further locates abnormal points. Besides, this model can ensure that future information cannot be used to predict the past, which benefits from the TCN network introducing causal convolution that the output at the current time step originates from the convolution of current as well as historical information. The authors conducted evaluations using OTIDS datasets in terms of accuracy, FPR, and TPR.

Luo et al. (2023), *PeerJ Comput. Sci.*, DOI 10.7717/peerj-cs.1648

**Table 11  Properties of hybrid DL-based IDS schemes.**

| Scheme | Protocol | Input features | Dataset | Effectiiveness (%) | | B/M[*] | Detect time | Methodology |
|--------|----------|----------------|---------|----------|-----|--------|-------------|-------------|
| | | | | Accuracy | F1 | | | |
| *Li et al. (2020)* | CAN | Payload | Self-Collected | 96 | – | B | – | the sparse auto-encoder for dimension reduction of features and RNN for classification. |
| *Nichelini et al. (2023)* | CAN | CAN ID & Payload | car-hacking dataset | 99 | 99 | B | CPU i7-8700K and GeForce GTX 1080 GPU: 0.2568ms | a modular framework including RNN autoencoder-based module |
| *Hanselmann et al. (2020)* | CAN | Payload | SynCAN & Self-Collected | 99 | – | B | – | LSTM per ID and autoencoder |
| *Lo et al. (2022)* | CAN | CAN ID & Payload | car-hacking dataset | 99.98 | 99 | B | – | use CNN and LSTM |
| *Agrawal et al. (2022)* | CAN | CAN ID & Payload | car-hacking dataset | – | 99 | B | Jetson Nano development board: 128.73 ms | use CNN and LSTM |
| *Song & Kim (2021)* | CAN | CAN ID | car-hacking dataset | 95.37 | 94.5 | B | – | use CNN and LSTM |
| *Sun et al. (2021)* | CAN | Payload | CAN Signal Extraction and Translation Dataset | – | 93.8 | B | Xavier with 8-core CPU: 5.7 ms | use CNN and Bi-LSTM with attention mechanism |
| *Javed et al. (2021)* | CAN | Payload | OTIDS | 95.09 | 93.79 | B | – | a combination of CNN and attention-based GRU model |

**Notes.**
[*]"B" and "M" refer to binary-classification and multi-classification, respectively.

Experimental results indicate that the proposed model performs better in fuzzing and DoS attacks. Also, *Thiruloga, Kukkala & Pasricha (2022)* presented TENET, an anomaly detection scheme that learns payload-level characteristics using TCN with the integrated attention mechanism.

**Complex valued neural network (CVNN):** *Han, Cheng & Ma (2021)* proposed an in-vehicle anomaly detection scheme using the CVNN combined with an encoder and attention mechanisms. Benefiting from the progress of CVNN in model safety, this model can alleviate the privacy disclosure problem caused by massive information collection for training deep learning-based models. The authors utilize a WGAN-based encoder to encrypt intermediate-layer features, followed by a CVNN analysis of in-depth features. The corresponding experimental results demonstrate that this model can protect the IDS while achieving 98% accuracy, and adversaries cannot extrapolate input information from the various features obtained. However, the model efficiency is at average levels compared to other DL-based schemes.

**Transfer learning:** *Mehedi et al. (2021)* presented a deep transfer learning-based P-LeNet method for IVN in order to improve the performance of learners on target domains by transferring the knowledge contained in source domains to the target domains. The experimental performance reveals that this model achieved an overall accuracy of 98.1%, which is higher than the other multiple traditional machine learning, deep learning, and deep transfer learning models. In addition, *Wei et al. (2022a)* proposed a domain adversarial neural network-based IDS consisting of a feature extractor, a label predictor, and a domain classifier. This model learns common feature representations from different but related domains, enabling it can learn the essential features of attacks to cope with the emergence of variant attacks. Furthermore, *Tariq et al. (2020)* introduced CANTransfer, a convolutional LSTM-based intrusion detection scheme for CAN bus using transfer learning technology. The properties of these IDS models are indicated in Table 12.

### Summary

Deep learning has great deployment potential in securing the in-vehicle network. In particular, the detection accuracy of abnormal traffic has reached a high level. For example, several schemes (*Kukkala, Thiruloga & Pasricha, 2020*; *Tariq, Lee & Woo, 2020*; *Desta et al., 2022*) have achieved close to 100% accuracy. However, the classification accuracy and computational complexity of intrusion detection models are in a trade-off relationship. The model with higher inference delay and hardware resource consumption is considered more complex, which negatively impacts the application of an IDS in resource-constrained environments in reality. The urgent challenge to be solved for the DL-based IDSs is reducing their inference delay and hardware resource consumption if the available computing power is poor or an IDS runs on a limited embedded environment of IVNs. In general, each deep learning technique has its own advantages and limitations when applied to the field of intrusion detection issues, as summarised in Table 13. The application of AEs, as an unsupervised learning technique, to intrusion detection schemes is generally used to reduce traffic data dimensions and filter noise. Furthermore, it is also used as a classifier by comparing the deviation between the reconstruction error and the preset threshold, that

Luo et al. (2023), *PeerJ Comput. Sci.*, DOI 10.7717/peerj-cs.1648

**Table 12  Properties of other DL-based IDS schemes.**

| Scheme | Protocol | Input features | Dataset | Effectiiveness (%) | | B/M[*] | Detect time | Methodology |
|---|---|---|---|---|---|---|---|---|
| | | | | Accuracy | F1 | | | |
| *Shi et al. (2021)* | CAN | CAN ID | OTIDS | 94.6 | – | B | – | TCN with parameterized Relu activation function |
| *Thiruloga, Kukkala & Pasricha (2022)* | CAN | Payload | SynCAN | – | – | B | Jetson TX2 with dual-core ARM Cortex-A57 CPUs: 0.25 ms | TCN with an integrated attention mechanism |
| *Han, Cheng & Ma (2021)* | CAN | CAN ID | Self-Collected | 99.3 | 84.3 | B | Intel i7-9500H CPU 3.60 GHZ and GTX 1650: 141 ms | CVNN combined with an encoder and attention mechanisms |
| *Mehedi et al. (2021)* | CAN | CAN ID & Payload | Car Hacking: Attack & Defense Challenge 2020 Dataset | 98.1 | 97.83 | B | – | a deep transfer learning-based LeNet model |
| *Wei et al. (2022a)* | CAN | Payload | Self-Collected | – | 99.6 | B | – | a CAN bus IDS based on a domain adversarial neural network |
| *Tariq et al. (2020)* | CAN | CAN ID & Payload | Self-Collected | – | 88.47 | B | – | a Convolutional LSTM based model using Transfer Learning |

**Notes.**

*"B" and "M" refer to binary-classification and multi-classification, respectively.

**Table 13 Deep learning technique applied to In-vehicle network IDS: key advantages and limitations.**

| Deep learning technique | Key advantages | Limitations |
|---|---|---|
| AE | The AE is a technique that helps reduce traffic data dimensions and filter noise in an unsupervised way. | The training process takes a long time. It can efficiently compress samples but is only available for similar ones. |
| RNN | RNN achieves adequate performance with historical memory and is an excellent tool for processing temporal sequence data. | It encounters the problem of gradient vanishing and exploding as the error accumulates over long terms. It requires high memory. |
| CNN | CNN is known for its properties of fewer network parameters. | It pertains to supervised learning and requires a large amount of labeled data. The pooling layer may lose valuable information. Models rely more on GPU, which is unsuitable for deployment on resource-constrained devices. |
| GAN | GAN is able to construct samples and, therefore, can alleviate the problem of imbalanced samples. | Training the GAN model is a challenge. The construct samples are probably unable entirely represent the real ones. Hence additional operations are required to check their validity. |
| DNN | DNN has the high abstraction ability to learn complex nonlinear relationships. | DNN requires mass sample data when the hidden layers increase. |
| DBN | DBN helps to solve the training problem of ANN. | A large number of network variables causes a long initialization phase. |

is, the input sequence is considered abnormal when the error is greater than the threshold. Although it can efficiently compress samples feature but is only available for similar ones and requires a long training time. In contrast, RNNs are an excellent tool since its adequate historical memory for processing temporal sequence data. However, it encounters the problem of gradient vanishing and exploding as the error accumulates over long terms. While different variants like LSTM and GRU are presented to address these issues, it also suffers from overfitting and high memory consumption. CNNs are known for their powerful image classification capabilities. When using CNNs to construct IDS, we need to consider the hardware resources of the on-board equipment since CNN-based models rely more on the computing power provided by the GPUs. Also, it pertains to supervised learning and requires a large amount of labeled data. The main contribution of GANs is that they can alleviate the problem of imbalanced samples by generating synthetic ones, but they also introduce additional operations to check their validity. Moreover, training a GAN model is a challenging task. As for DNNs, they have the high abstraction ability to learn complex nonlinear relationships, but with the increase of hidden layers, a large amount of sample data is required to train the network parameters. Another deep learning technique that can be used to design intrusion detection models is DBNs which help to initialize network parameters. Also, it needs a long initialization phase.

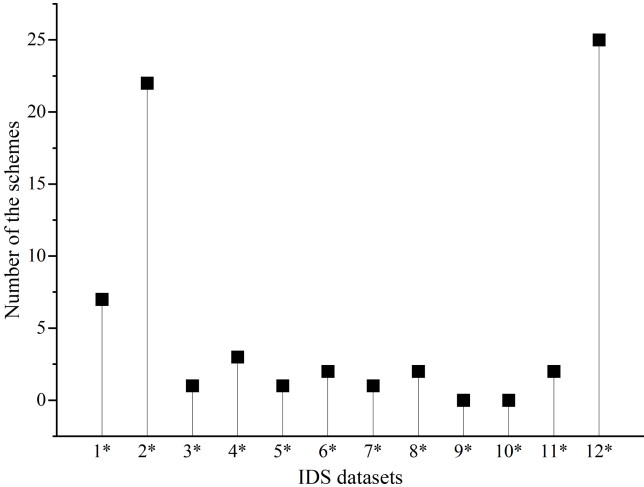

**Figure 7  Datasets applied in the DL-based IDS schemes for IVN.** The corresponding relationship between the $X$-axis numbers and the dataset names is as follows: $1^*$: OTIDS; $2^*$: Car Hacking Dataset; $3^*$: Survival Analysis Dataset; $4^*$: SynCAN; $5^*$: CAN Signal Extraction and Translation Dataset; $6^*$: Attack & Defense Challenge 2020 Dataset; $7^*$: SOME/IP Dataset; $8^*$: Automotive Ethernet Intrusion Dataset; $9^*$: Automotive CAN Bus Intrusion Dataset v2; $10^*$: CAN Log Infector & Ambient CAN Traces; $11^*$: ROAD Dataset; $12^*$: Self-Collected.

## DISCUSSION

This section provides a discussion and insights about the DL-based IDS schemes investigated in the previous section to researchers who are interested in applying deep learning algorithms to solve cybersecurity threats.

### Dataset

Using a proper dataset is an essential prerequisite that constructs an efficient and effective IDS. we surveyed the benchmark datasets used by IDS for training, validating, and testing. As exhibited in Fig. 7, the majority of proposed IDS used the self-collected dataset where the data is inaccessible, while the car hacking dataset is the most commonly used public benchmark. However, according to the generation procedure of the car hacking dataset described by *Seo, Song & Kim (2018)*, the payloads of the attack messages are randomly synthesized and periodically transmitted at a high frequency, which may be extremely different from real-world attacks. An experienced adversary may not perform an attack with random payloads but instead pad the payloads with similar signal values, making them look like normal messages (*Zhao et al., 2022*). Additionally, several IDS schemes are verified with two datasets, which may increase the generalization capability of the model.

As shown in Table 14, the distribution of traffic samples across the normal and attack classes is imbalanced since network intrusion behaviors are uncommon. The capacity to handle imbalanced datasets is an important research content. Some of the surveyed schemes utilize the DL network to generate synthetic samples for classes with relatively little data. These methods usually employ GAN-based models to generate synthetic samples. Because GAN-based models can extract the latent distribution of the original data as well

**Table 14  The attack type and data size of car hacking dataset.**

| Attack type | Overall messages | Normal messages (%) | Injection messages (%) |
|---|---|---|---|
| DoS | 3,665,771 | 84.0% | 16.0% |
| Fuzzy attack | 3,838,860 | 87.2% | 12.8% |
| Spoofing (drive gear) | 4,443,142 | 86.6% | 13.4% |
| Spoofing (RPM gauze) | 4,621,702 | 85.8% | 14.2% |

as generate artificial samples of unknown attacks, increasing samples to balance the dataset. In addition, it can avoid the problems of losing information and generating redundant samples existing in undersampling or oversampling methods. But the synthetic data is probably unable completely represent the real traffic, and additional operations need to be considered to check its validity.

### DL role in IDS

We review the articles published between 2016 and the first half of 2023 on DL-based IDS schemes for IVN. Figure 8 outlines the frequency of deep learning methods in the investigated articles to explore the model selection preferences of researchers when detecting the attacks in the vehicular network. As shown in this figure, More than a quarter of the articles have used RNNs and variants (*e.g.*, LSTM) to construct the detector, while DBMs are the least applied model overall. The figure also reveals that 13% of authors convert network traffic into images and use CNNs due to their capability to extract advanced feature representations from spatial data, and other authors considered the communication traffics as time series data. Furthermore, we found that about 16% of IDS schemes benefited from two DL networks to deal with diverse attacks and anomalies, which were classified as hybrid schemes. The combination of CNNs and RNNs is used most in hybrid-based IDS schemes, while the AE+RNN is also applied in several schemes.

DL-based IDS schemes achieved high performance by employing different network architectures. We compared the effectiveness of different methods relying on the same benchmark datasets. Figure 9 indicates the accuracy of the analyzed articles that employed the car hacking dataset to evaluate their model. However, we underline that it is not a comprehensive comparison as some papers did not use accuracy as their model evaluation criterion and are therefore not included here. Besides, The experimental results we collected were obtained from isolated studies by different researchers based on diverse experimental aspects, such as different hyperparameters, optimizers, and neuron numbers, which all affect the detection performance of the model.

### Feature selection

We analyze which traffic fields are most favored when designing detectors in the reviewed IDS schemes. Based on this purpose, Fig. 10 indicates the percentage of the IDS models applying different traffic fields. As shown in the figure, the fields selected by the CAN bus IDS scheme can be divided into CAN ID only, payload only, and the combination of both CAN ID and payload, while the vehicle Ethernet scheme mainly uses the protocol

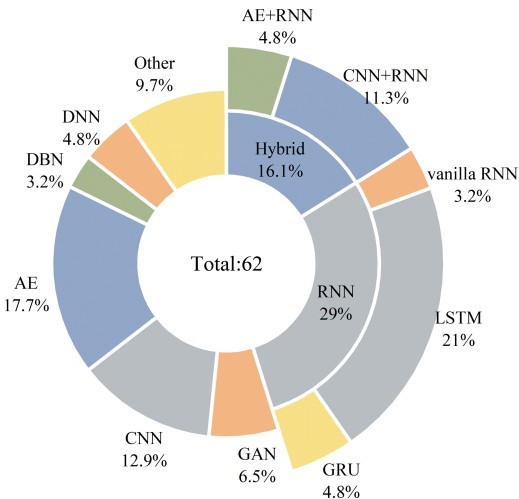

**Figure 8** **Use of DL models in the surveyed schemes.**

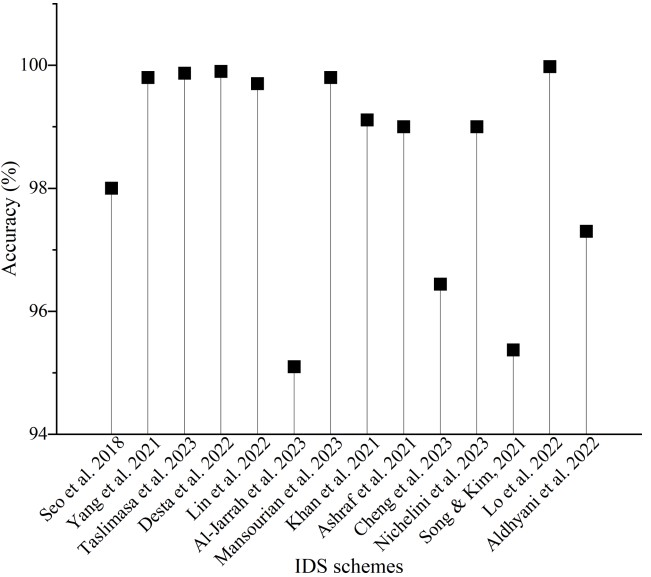

**Figure 9** **Accuracy of the studied schemes on the car hacking dataset.**

header because the maximum byte of the payload reaches 1,500. Around one-quarter of the IDS approaches only use CAN ID as the input feature of the DL network. These methods can detect malicious injection attacks, such as DoS, fuzzing, spoofing, and replay attacks, benefiting from the periodic transmission mode of CAN traffic. However, they are limited to detecting packet-level anomalies (as opposed to signal-level anomalies) that cause changes in packet order or invalid CAN ID rather than signal tampering, which means they are incapable of identifying masquerade attacks and cannot take advantage of the dependencies between different signals. About 68% of the papers have used the payload

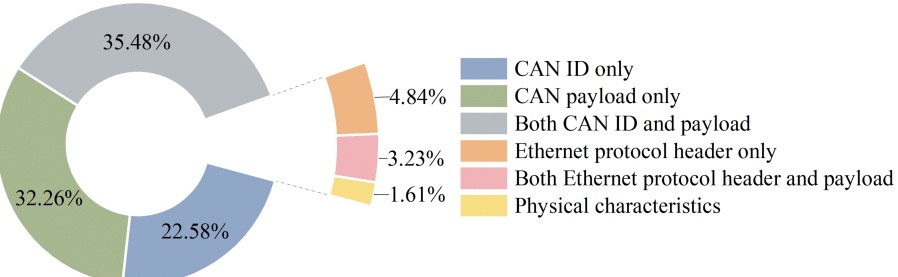

**Figure 10** Use of traffic field in the surveyed papers.

field (including payload only, and both CAN ID and payload), enabling the model to recognize payload-based masquerade attacks, such as abnormal changes in specific signal values. It deserves further mention that which fields are selected as model input features depends on the attack scenarios anticipated to be defended.

## FUTURE RESEARCH DIRECTIONS

Using deep learning technology to develop intrusion detection systems for IVNs has received extensive attention in recent years. Unfortunately, they encounter challenges across multiple dimensions, including limited on-board hardware resources, inappropriate datasets, larger inference latency, and high false alarm rates. This section presents some lessons learned and critical directions for future research, intending to promote the application of the DL-based IDS in reality.

**Apply hybrid schemes:** False alarms in in-vehicle cybersecurity scenarios can be extremely costly. Considering the large network traffic scale, even a low false alarm rate may consume much maintenance personnel time. Assume a detector achieves 90% accuracy in the worst case and traffics is transmitted at 10ms intervals. This means ten messages cannot be correctly identified within one second, seriously affecting subsequent protection decisions. For this, A practical intrusion detector does not rely solely on a single technique but is an ensemble of effective components. Researchers could deploy post-processing solutions to avoid false alarms caused by factors such as traffic jitter in a short period, like calculating confidence scores for detect results. Besides, utilizing both temporal and spatial features of traffics may benefit extracting more latent sample characteristics and then improve the detection performance of IDSs. The upcoming studies can further focus on hybrid detection schemes in detecting known and unknown attacks.

**Use hardware accelerators:** Most proposed IDSs are based on the principle of offline training and online detection. The inference process in a limited embedded resource environment is more concerned than the training process since the inference process is performed on-board while the training process is offline. Generally, the inference process consists of preprocessing received traffics and feature computation, which will be impacted by computing power and allocated memory during IDS deployment. Therefore, in the following research on the DL-based intrusion detection field, leveraging hardware accelerators or offloading tasks could be feasible methods to reduce inference time while

maintaining advanced IDS solutions. With the widespread application of deep learning, chip vendors have produced advanced AI accelerators such as field programmable gate array (FPGA), application-specific integrated circuit (ASIC), neural processing unit (NPU), and edge TPU. Although the investigated DL-based IDS schemes only rely on CPUs and GPUs, there are some literature works (*Khandelwal & Shreejith, 2022*; *Khandelwal, Wadhwa & Shreejith, 2022*; *Zhang, Yan & Ma, 2022*; *Machupalli, Hossain & Mandal, 2022*) that have applied these chips for training neural networks.

**Secure next-generation IVN protocol:** Currently, the great majority of the reviewed IDS articles are designed for the CAN bus. However, Automotive Ethernet is gradually expanding its application domain in-vehicle networks due to advances in bandwidth, cost, built-in security, and native support for TCP/IP. Its practical applications so far mainly include DoIP, AVB, SOME/IP, and TSN. The introduction of Ethernet makes it possible to attack utilizing legacy Internet attacks in automotive devices directly. Unlike the CAN bus communication method, the Automotive Ethernet belongs to point-to-point communication, with a longer protocol header and numerous upper-layer protocols. These lead to the IDSs designed for CAN cannot be fully adapted to Ethernet. Therefore, the IDS solutions for automotive Ethernet Protocol Suite should be considered in future studies to secure next-generation IVNs.

**Develop assessment method:** Academic research has difficulties comprehensively evaluating the benefits of deep learning models. The comparison among different IDS schemes is conducted on diverse experimental aspects, which cannot provide an impartial result in terms of efficiency and effectiveness. This is owing to the diversity of the employed dataset, the adopted dataset segment, pre-processing process, neuron arrangement, and hardware platforms. Consequently, it is necessary to have a complete range of verification platforms for automotive intrusion detection system performance, which can obtain fairer comparison results under a unified computing resource and common influencing factors. For instance, *Stachowski, Gaynier & LeBlanc (2019)* developed a test suite to assess the performance of automotive IDS products. And *Wang et al. (2022)* make horizontal comparison analyses of ten representative IDSs.

## CONCLUSION

A practical IDS should have extensive attack detection capabilities, low false-positive rate, and high confidence in detection. In order to improve the effectiveness of IDS against new security challenges, numerous researchers have intended to incorporate various deep learning techniques to enhance the feature extraction and classification steps. Therefore, there are many DL-based IDS schemes have been proposed and designed in recent years, and this survey aims to review and categorize them extensively. Through a systematic review, we briefly illustrated a variety of attacks on IVNs, and their possible countermeasures that are best suited to detect them; in addition, the major datasets that benefited to analyze and evaluate the IDS solutions are presented. To comprehensively offer in-depth knowledge about the recent research status of deep IDS methods, we put forward a taxonomy of the proposed IDS approaches concerning their employed neuron network architecture.

Then, we provide a detailed discussion under each category by analyzing their input data strategy, benchmark datasets, classification technique, and evaluation criteria. The current findings demonstrate that no matter practicability or security, no perfect IDS solution can effectively discover all potential attacks. Thus, this work will be conducive to enhancing the understanding of current deep IDS studies on IVNs and searching for new and more suitable techniques.

### Funding
This work was supported by the Shanghai Pudong New Area Science and Technology Development Fund, Industry-University-Research Special Project (Future Vehicle) (No. PKX2022-W01). The funders had no role in study design, data collection and analysis, decision to publish, or preparation of the manuscript.

### Grant Disclosures
The following grant information was disclosed by the authors:
Shanghai Pudong New Area Science and Technology Development Fund, Industry-University-Research Special Project (Future Vehicle): PKX2022-W01.

### Competing Interests
The authors declare there are no competing interests.

### Author Contributions
- Feng Luo conceived and designed the experiments, analyzed the data, authored or reviewed drafts of the article, and approved the final draft.
- Jiajia Wang conceived and designed the experiments, analyzed the data, prepared figures and/or tables, authored or reviewed drafts of the article, and approved the final draft.
- Xuan Zhang performed the experiments, analyzed the data, authored or reviewed drafts of the article, and approved the final draft.
- Yifan Jiang performed the experiments, authored or reviewed drafts of the article, and approved the final draft.
- Zhihao Li performed the computation work, prepared figures and/or tables, and approved the final draft.
- Cheng Luo performed the computation work, prepared figures and/or tables, and approved the final draft.

### Data Availability
This is a literature review and did not utilize raw data.

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
