# Peer review of "In-vehicle network intrusion detection systems: a systematic survey of deep learning-based approaches"

_PeerJ Computer Science, doi:10.7717/peerj-cs.1648_

## Round 0.1 · original submission · Minor Revisions

The reviewers have made some recommendations on issues to be addressed for the revised manuscript.

**Language Note:** PeerJ staff have identified that the English language needs to be improved. When you prepare your next revision, please either (i) have a colleague who is proficient in English and familiar with the subject matter review your manuscript, or (ii) contact a professional editing service to review your manuscript. PeerJ can provide language editing services - you can contact us at copyediting@peerj.com for pricing (be sure to provide your manuscript number and title). – PeerJ Staff

Reviewer 1 ·

Basic reporting

The article first gives some descriptions of IDS for vehicles that can be found in several related surveys. After this, the article has an adequate introduction to the subject, and the authors make it clear what they expect from this study. The section on background knowledge describes in detail how IDS schemes are trained, validated, and tested using datasets. The article offers sufficient information about networks in vehicles that are vulnerable to a variety of cyberattacks. Figure 6 helps the readers to understand the high-level ideas of different deep-learning network architectures, such as RNN, CNN, GAN, and DNN.

Experimental design

This is survey research. In this article, the authors use a systematic literature review process (planning, conducting, and reporting) to collect more detailed articles on the deep learning techniques designed by the IDS. The five RQ (Research Questions) defined in the planning stage help to organize the overall article structure more straightforwardly. The use of snowballing process offers a thorough examination of related research works targeting on intrusion detection, anomaly detection, and in-vehicle network. Also, it's great that this article provides a comprehensive collection of deep learning-based IDS schemes published between 2016 and 2022.

Validity of the findings

As a result of this work, one major novelty is that the authors present a fine-grained taxonomy based on the neural network architectures in order to classify the state-of-the-art DL-IDS schemes on their capabilities. The article compares and critically examines the investigated solutions by examining their methods, datasets, and evaluation metrics. At the end of the article, the authors identify possible future research directions for improving DL-based IDS performance. The conclusion part is appropriately stated and connects to the original question investigated.

Additional comments

Please list more existing methods based on traditional machine learning that cannot handle the security risks very well.

Please add more references to some statements made in the article, such as the significant advantages of IVN IDSs, and the capability of deep learning networks in identifying sophisticated attacks and zero-day attacks.

Please explain more about why the IDS schemes are limited to detecting packet-level anomalies.

Have you deeply investigated the usage of a combination of more than one DL network? The article mentioned that the combination of CNNs and RNNs is used most in hybrid-based IDS schemes, are there any other combinations used in any specific schemes?

Reviewer 2 ·

Basic reporting

no comment

Experimental design

no comment

Validity of the findings

no comment

Additional comments

This paper gives a survey on deep learning-based in-vehicle network intrusion detection methods. It covers many recent approaches and performs some comparison. The paper was well-written. Some problems need to be solved to improve the quality of the paper.
(1) It is suggested to cover more literature published in 2023, so as to increase the impact of the paper.
(2) The paper can show some comparison experimental results on public benchmark datasets and conduct analysis.
(3) The size of different datasets can be shown.
(4) The future directions can be organized in a more structural manner to highlight different topics.

---

## Round 0.2 · accepted · Accept

The original Academic Editor is unavailable so I am taking over handling the submission in my capacity as Section Editor.

We are happy to inform you that your manuscript has been accepted for publication.

Reviewer 1 ·

Basic reporting

The article first gives some descriptions of IDS for vehicles that can be found in several related surveys. After this, the article has an adequate introduction to the subject, and the authors make it clear what they expect from this study. The section on background knowledge describes in detail how IDS schemes are trained, validated, and tested using datasets. The article offers sufficient information about networks in vehicles that are vulnerable to a variety of cyberattacks. Figure 6 helps the readers to understand the high-level ideas of different deep-learning network architectures, such as RNN, CNN, GAN, and DNN.

Experimental design

This is survey research. In this article, the authors use a systematic literature review process (planning, conducting, and reporting) to collect more detailed articles on the deep learning techniques designed by the IDS. The five RQ (Research Questions) defined in the planning stage help to organize the overall article structure more straightforwardly. The use of snowballing process offers a thorough examination of related research works targeting on intrusion detection, anomaly detection, and in-vehicle network. Also, it's great that this article provides a comprehensive collection of deep learning-based IDS schemes published between 2016 and 2022.

Validity of the findings

As a result of this work, one major novelty is that the authors present a fine-grained taxonomy based on the neural network architectures in order to classify the state-of-the-art DL-IDS schemes on their capabilities. The article compares and critically examines the investigated solutions by examining their methods, datasets, and evaluation metrics. At the end of the article, the authors identify possible future research directions for improving DL-based IDS performance. The conclusion part is appropriately stated and connects to the original question investigated.

Additional comments

The revised version addressed all my questions/concerns in the first round of review.

Reviewer 2 ·

Basic reporting

no comment

Experimental design

no comment

Validity of the findings

no comment

Additional comments

The authors have addressed all my concerns.